# ABC Transporter C1 Prevents Dimethyl Fumarate from Targeting Alzheimer’s Disease

**DOI:** 10.3390/biology12070932

**Published:** 2023-06-29

**Authors:** Luisa Möhle, Katja Stefan, Pablo Bascuñana, Mirjam Brackhan, Thomas Brüning, Ivan Eiriz, Ahmed El Menuawy, Sylvie van Genderen, Irene Santos-García, Anna Maria Górska, María Villa, Jingyun Wu, Sven Marcel Stefan, Jens Pahnke

**Affiliations:** 1Department of Pathology, Section of Neuropathology/Translational Neurodegeneration Research and Neuropathology Lab, University of Oslo (UiO) and Oslo University Hospital (OUS), Sognsvannsveien 20, 0372 Oslo, Norway; 2Pahnke Lab (Drug Development and Chemical Biology), Lübeck Institute of Experimental Dermatology (LIED), University of Lübeck (UzL) and University Medical Center Schleswig-Holstein (UKSH), Ratzeburger Allee 160, 23538 Lübeck, Germany; 3School of Medical Sciences, Faculty of Medicine and Health, The University of Sydney, Camperdown, NSW 2006, Australia; 4Department of Pharmacology, Faculty of Medicine, University of Latvia, Jelgavas iela 3, 1004 Rīga, Latvia; 5Department of Neurobiology, The Georg S. Wise Faculty of Life Sciences, Tel Aviv University, Tel Aviv 6997801, Israel

**Keywords:** dimethyl fumarate, DMF, Alzheimer’s disease, AD mouse model, drug repurposing, multiple sclerosis, ABC transporter, ABCC1, blood–brain barrier, fingolimod, BBB

## Abstract

**Simple Summary:**

Reusing drugs could potentially shorten the development time for the effective treatment of dementia. Here, we tested a drug—dimethyl fumarate—for its efficacy in reducing Alzheimer’s disease-related changes in the brain. We discovered that the drug is not efficient due to the involvement of an essential exporting brain transporter.

**Abstract:**

Alzheimer’s disease (AD), the leading cause of dementia, is a growing health issue with very limited treatment options. To meet the need for novel therapeutics, existing drugs with additional preferred pharmacological profiles could be recruited. This strategy is known as ‘drug repurposing’. Here, we describe dimethyl fumarate (DMF), a drug approved to treat multiple sclerosis (MS), to be tested as a candidate for other brain diseases. We used an APP-transgenic model (APPtg) of senile β-amyloidosis mice to further investigate the potential of DMF as a novel AD therapeutic. We treated male and female APPtg mice through drinking water at late stages of β-amyloid (Aβ) deposition. We found that DMF treatment did not result in modulating effects on Aβ deposition at this stage. Interestingly, we found that glutathione-modified DMF interacts with the ATP-binding cassette transporter ABCC1, an important gatekeeper at the blood–brain and blood–plexus barriers and a key player for Aβ export from the brain. Our findings suggest that ABCC1 prevents the effects of DMF, which makes DMF unsuitable as a novel therapeutic drug against AD. The discovered effects of ABCC1 also have implications for DMF treatment of multiple sclerosis.

## 1. Introduction

### 1.1. Alzheimer’s Disease: Background and Therapy

Approximately 55 million people worldwide suffer from dementia, and the numbers are estimated to rise to over 130 million by 2050 [1,2,3]. The most common cause of dementia is Alzheimer’s disease (AD) [1,2]. Two very well-recognized histopathological hallmarks of AD are (i) neurofibrillary tangles (NFT) consisting of aggregates of the hyperphosphorylated microtubule-associated protein τ, and (ii) the so-called ‘senile plaques’ of agglomerated amyloid-β (Aβ) polypeptides with a length between 37 and 43 amino acids. The Aβ peptides are produced mainly by β- and γ-secretase cleavage of the β-amyloid-precursor protein (APP) [4,5]. Permanent production of Aβ results in toxic aggregates of both NFT and Aβ, causing neurodegeneration through impairment of neurite function, disrupted synaptic communication, and disturbed cortico–cortical circuits (reviewed in [6]). Clinically, this leads to well-described symptoms such as cognitive impairment with memory loss, disorientation, agnosia, and also neuropsychiatric abnormalities [7,8].

Despite more than a century of research efforts, the treatment options for AD are limited to very few symptomatic therapeutics, addressing either (i) dementia-related symptoms, such as cognitive decline, or (ii) secondary symptoms, such as depression [9]. Only four therapeutics are currently licensed on the European and US markets to treat AD-related symptoms (Figure 1): the acetylcholine esterase (AChE) inhibitors donepezil (**1**), galantamine (**2**), and rivastigmine (**3**), as well as the *N*-methyl-_D_-aspartate (NMDA) receptor antagonist memantine (**4**).

The first approved drug to treat AD, the AChE inhibitor tacrine (**5**) [10], has already been withdrawn from the markets due to doubtful benefits and adverse side effects [11].

In an attempt to establish a causative treatment, the γ-secretases inhibitor semagacestat (**6**) has been clinically evaluated. However, the phase III trial was terminated due to severe side effects [12].

### 1.2. Drug Repurposing

Drug repurposing is the exploration of novel therapeutic indications for drugs already in clinical use. It has considerable advantages in modern drug development: (i) it extends the knowledge with respect to the polypharmacological profile of the drugs in terms of their initial indication, increasing data awareness and the general safety profile; (ii) due to the known and well-documented safety profile, new clinical applications can be accomplished much faster; and (iii) as these drugs have already gone through the approval process, their application to new indications is associated with considerably lower costs than drugs from novel drug discovery pipelines that have to undergo investigations in all phases. Furthermore, drug repurposing to treat neurodegenerative diseases is highly encouraged by the European Medical Agency (EMA, Amsterdam, Nederlands) and the Federal Drug Agency (FDA, Silver Springs, MD, USA). Therefore, mutual recognition agreements have been established between several countries (https://www.ema.europa.eu/en/human-regulatory/research-development/compliance/good-manufacturing-practice/mutual-recognition-agreements-mra (accessed on 10 May 2023)).

### 1.3. Dimethyl Fumarate

A potential candidate for repurposing strategies for AD is DMF (**7**; Tecfidera^®^, Skilarence^®^). DMF and its bioactive metabolite monomethyl fumarate (MMF, **8**; Bafiertam^®^) are approved for the treatment of multiple sclerosis (MS) [13], an inflammatory brain disease, and psoriasis, a disease of the skin (reviewed in [14]). DMF regulates inflammation [15,16,17], halts disease progression in MS [18,19,20,21], and affects the response to oxidative stress [22]. Several studies in animal models have previously reported beneficial effects of DMF treatment on 1-methyl-4-phenyl-1,2,3,6-tetrahydropyridine-(MPTP)- or Aβ-mediated neurotoxicity [23,24,25], stroke, intracerebral hemorrhage [26,27], as well as learning and memory in streptozotocin-induced rat models [28,29,30]. DMF may also induce ATP-binding cassette (ABC) transporters [31,32], a protein class that has been associated with direct and/or indirect Aβ clearance from the brain in several independent studies [33,34]. Figure 2 shows the molecular formulae of DMF and MMF, including the important physicochemical parameters calculated octanol–water partition coefficient (CLogP), molecular weight (MW), molar refractivity (MR), and topological polar surface area (TPSA) [35] as determined with the online web service SwissADME [36].

In contrast, our own study in young female APPtg mice found no effects of DMF treatment on cognitive performance, the extent of β-amyloidosis, or neuroinflammation markers [3]. These results prompted us to evaluate DMF at more advanced pathological stages of β-amyloidosis in this model, as several differences between early- and late-stage amyloidosis mice exist (e.g., the integrity of BBB, the regulation and function of BBB-located proteins, or altered CNS penetration of drugs and/or metabolites amongst other factors). In addition, we explored potential options for the (in)effectiveness of DMF, which are presented in the current work.

## 2. Material and Methods

### 2.1. Animal Models and Breeding Scheme

Heterozygous female and male APPPS1-21 mice [B6.Cg-Tg(Thy1-APPSw,Thy1-PSEN1*L166P)/21JkcrPahnk, APPtg [37]] were housed in Eurostandard type III cages (macrolone) in groups of 5–6 animals per cage at the animal core facility of the Department of Comparative Medicine (section Radium Hospital) at the Oslo University Hospital (Norway) with a 12 h/12 h light/dark cycle and free access to food (Rat and Mouse No.1 Maintenance expanded pellets from SDS) and water (pH 3 for maintenance, pH 7.2 for treatment with DMF) at a mean temperature of 22 °C [38]. All cages were provided with aspen wood (*Populus tremula*, Tapvei, Estonia) as bedding substrate and additional enrichment material (tissue paper, tunnels/huts, and occasionally gnawing sticks). APPPS1-21 mice have a combined *APP* (Swedish mutations) and *PS1* (L166P mutation) transgene under the control of the *Thy1*-promoter, leading primarily to pathological Aβ production in the fronto-cortical neurons and the first cortical Aβ plaques at 45–50 days of age, which also occurs much later in other brain regions, but to a significantly lesser extent (e.g., hippocampus) [34,39]. All experiments were conducted in accordance with the guidelines for animal experiments of the European Union directive and national laws.

### 2.2. Treatment Scheme

Animals were treated with dimethyl fumarate (DMF (97% purity, Merck, Darmstadt, Germany)) in drinking water (pH ~7.2) for 50 days from 125 to 175 days of age. DMF was dissolved in tap water by stirring it for 1–2 h at a slightly elevated temperature (ca. 25–35 °C). Neutral pH was important to avoid the degradation of DMF [40]. Once DMF was dissolved, water was filtered sterile. Fresh, sterile water with DMF was provided once per week, and DMF concentration was adjusted throughout the experiment to achieve a daily uptake of 75 mg/kg DMF. To this end, body weight and water consumption were monitored weekly throughout the entire experiment. Control groups received sterile filtered water at neutral pH (~7.2).

### 2.3. Spectroscopic Measurement of DMF

We followed a previously described method using UV spectroscopy [41]. We prepared a reference curve by dissolving DMF in tap water with neutral pH of 1 mg/mL. From this stock, several dilutions were prepared spanning 0.002—1 mg/mL, and absorbance at 210 nm was measured using a NanoDrop™ One UV-Vis spectrophotometer (Thermo Scientific, Schwerte, Germany). There was a strong linear correlation between absorbance and DMF concentration from 0.002 to 0.1 mg/mL (Appendix B, Figure A1).

To assess the degradation of DMF in water, we prepared solutions at the relevant concentration of 0.6 mg/mL. Absorbance was measured in freshly prepared solutions as well as in aliquots kept for 1–2 weeks either at room temperature or in the fridge. Prior to measurement, samples were diluted 10-fold.

### 2.4. Tissue Harvesting

Mice were euthanized using ketamine/xylazine (400 mg/kg ketamine, 40 mg/kg xylazine). After intracardial perfusion with ice-cold PBS, brains were removed and separated into two hemispheres. One hemisphere was kept in paraformaldehyde (PFA 4% in PBS), the other on snap frozen in liquid nitrogen and later transferred to −80 °C (for protein extraction).

### 2.5. Protein Extraction and Quantification

Frozen hemispheres were thawed on ice in 500 µL RNAlater^®^ (Merck KGaA, Germany) for one hour, removed from the liquid, and homogenized for 60 s with four 2.8 mm ceramic beads (OMNI International, Kennesaw, GA, USA) using the SpeedMill PLUS (Analytik Jena GmbH, Jena, Germany). Twenty mg of homogenate was mixed with 10 µL cold Tris-buffered saline (TBS, pH 7.5, containing protease inhibitor (Roche, Germany)) per 1 mg brain. Samples were homogenized with a 2.8 mm ceramic bead (SpeedMill PLUS, 30 s) and centrifuged (16,000× *g*, 4 °C, 20 min) to separate soluble and aggregated Aβ. The resulting supernatant (TBS fraction containing soluble Aβ) was collected and stored at −20 °C until further use. The pellet was mixed with an 8 µL cold 5 M guanidine buffer (pH 8.0) per 1 mg brain homogenate and homogenized (SpeedMill PLUS, 30 s). Samples were incubated at room temperature for 3 h under constant shaking (1500 rpm) before centrifugation (16,000× *g*, 4 °C, 20 min). The supernatant (guanidine fraction containing aggregated Aβ) was collected and stored at −20 °C until further use. To quantify Aβ42 in TBS and guanidine fractions, we performed electrochemiluminescence immunoassays using the V-PLEX Plus Aβ42 Peptide (4G8) Kit and a MESO QuickPlex SQ120 machine according to manufacturer’s recommendations (Meso Scale Diagnostics, Rockville, MD, USA).

### 2.6. ABC Transporter Assays: Cell Culture

The ABCB1-, ABCC1-, and ABCG2-expressing cell lines A2780/ADR, H69AR, and MDCK II BCRP were a generous gift of Prof. Dr. Finn K. Hansen and Prof. Dr. Gerd Bendas (Pharmaceutical and Cell Biological Chemistry, University of Bonn, Germany). Cells were cultured as previously described [42,43]. A2780/ADR and H69AR cells were cultivated using RPMI-1640 cell culture media (VWR, Norway) supplemented with 10% and 20% fetal bovine serum (FBS; VWR), respectively. MDCK II BCRP cells were cultured in DMEM cell culture media (VWR) supplemented with 10% FBS. All cells were also supplemented with streptomycin (50 µg/µL), penicillin G (50 U/mL), and L-glutamine (2 mM; VWR). The cells were stored under liquid nitrogen (media: 90%; DMSO: 10%, Alfa Aesar/Thermo Fisher Scientific, Oslo, Norway) and cultivated at 37 °C under a 5% CO_2_-humidified atmosphere. A trypsin-EDTA solution (0.05%/0.02%; VWR) was used to detach the cells for either sub-culturing or biological evaluation at a confluence of ~90%, followed by washing steps and the addition of fresh cell culture media. Cell counting was performed using a Scepter handheld automated cell counter (60 µM capillary sensor; MerckMillipore, Darmstadt, Germany).

### 2.7. ABC Transporter Assays: Membrane Preparation

Forty full-grown (confluence ≥ 90%) cell culture dishes (VWR) with ABCC1-expressing H69AR cells were necessary to obtain a membrane preparation with adequate protein (and therefore ABCC1) content. The cells were harvested using a freshly prepared homogenization buffer (4-(2-hydroxyethyl)-piperazin-1-ethansulfonic acid (HEPES; 20 mM; Alfa Aesar) and Na_2_-EDTA (10 mM; Sigma-Aldrich, Oslo, Norway)). After adding a 2.4 mL homogenization buffer to each dish, the cells were scratched from the bottom and transferred into 50 mL reaction tubes (VWR) on ice, subsequently repeating this step. The cell suspension was shredded three times, applying a dispenser (Polytron, Kinematica AG, Luzern, Switzerland). The homogenized suspension was ultracentrifuged (40,000× *g*, 10 min, 4 °C). The supernatant was disposed of in a freshly prepared storage buffer (HEPES (20 mM) and Na_2_-EDTA (0.1 mM)), subsequently followed by two further ultracentrifugation steps. The final membrane preparation was suspended in a storage buffer and aliquoted before storage at −80 °C. Protein content was determined by applying a protein content determination kit (Pierce™ BSA assay, Thermo Fisher Scientific, Norway; 7.0 mg/mL).

### 2.8. ABC Transporter Assays—Experimental Protocols

The assays to assess the ABC transporter activity were conducted as reported earlier [43,44,45]. We assessed the activity of ABCB1 (calcein AM and daunorubicin), ABCC1 (daunorubicin and rhodamine 123), and ABCG2 (pheophorbide A and Hoechst 33342), and the fluorescence probes were supplied by Calbiochem (Merck KGaA, Darmstadt, Germany), Biomol (Cayman Chemicals, Ann Arbor, MI, USA), Sigma–Aldrich, and MerckMillipore (Merck KGaA, Darmstadt, Germany).

DMF and MMF were prepared in a volume of 20 µL at concentrations of 100 µM or 500 µM in clear (calcein AM, daunorubicin, rhodamine 123, and pheophorbide A) or black (Hoechst 33342) 96-well flat-bottom plates (Brand, Germany). Next, we added a 160 µL cell suspension containing either 30,000 cells/well (calcein AM and Hoechst 33342) or 45,000 cells/well (daunorubicin, rhodamine 123, and pheophorbide A) in either phenol red-free RPMI-1640 (A2780/ADR and H69AR) or phenol red-free DMEM (MDCK II BCRP) without further supplements. DMF and MMF were incubated with the cells for 30 min before adding the respective fluorescence dye to each well (20 µL of calcein AM (3.125 µM), daunorubicin (30 µM), rhodamine 123 (3 µM), pheophorbide A (5 µM), or Hoechst 33342 (10 µM); final concentrations: calcein AM: 0.3125 µM; daunorubicin: 3 µM; rhodamine 123: 0.3 µM; pheophorbide A: 0.5 µM; Hoechst 33342: 1 µM). Subsequent fluorescence measurements depended on the dye:*Calcein AM*: Fluorescence (excitation: 485 nm; emission: 520 nm) was measured for 30 min at 30 s intervals using a Paradigm^®^ microplate reader (Beckman Coulter Biomaterials, Munich, Germany);*Daunorubicin*: Fluorescence (excitation: 488 nm; emission: 695/50 nm) was measured after 180 min incubation on an Attune NxT flow cytometer (Invitrogen, Waltham, MA, USA);*Rhodamine 123 and pheophorbide A*: Fluorescence (excitation: 488 nm; emission: 695/50 nm) was measured after 120 min incubation on an Attune NxT flow cytometer;*Hoechst 33342*: Fluorescence (excitation: 360 nm; emission: 460 nm) was measured after 120 min incubation using a Paradigm^®^ microplate reader.

The slopes (calcein AM) or average fluorescence values (daunorubicin, rhodamine 123, pheophorbide A, Hoechst 33342) per well were calculated and compared to the reference inhibitors cyclosporine A (ABCB1), 4-(4-(benzo[d][1,3]dioxol-5-ylmethyl)piperazin-1-yl)-6,7,8,9-tetrahydropyrimido [4,5-b]indolizine-10-carbonitrile (ABCC1) [46], and Ko143 (ABCG2). Data were processed using Prism (v9, GraphPad Software, San Diego, CA, USA). For full-blown concentration-effect curves, dilution series of both DMF and DMS-SG were generated in a concentration range between 0.05 µM and 100 µM. DMS-SG was generated according to a previously established protocol [47] by an equimolar mixture of DMF and glutathione (incubation: 30 min) and subsequent generation of the dilution series.

### 2.9. ABC Transporter C1 ATPase Assay

The vanadate-sensitive ATPase assay was performed as already described before [47,48,49,50] with minor modifications. The reaction mixture [3-(N-morpholino)propanesulfonic acid-(MOPS)-Tris (40 mM; pH 7.0; Sigma–Aldrich, Oslo, Norway), KCl (50 mM; Sigma-Aldrich), dithiothreitol (2 mM; Alfa Aesar), EGTA-Tris (500 µM; pH 7.0; Sigma–Aldrich), sodium azide (5 mM; Sigma–Aldrich), ouabain (1 mM; Alfa Aesar,) was supplemented with 10 µg of the ABCC1 membrane preparation (2 mg/mL). DMF, DMS-SG, or GSH in DMSO (final DMSO concentration < 1%) were added (20 µL). The reaction was started by the addition of MgATP (3.3 mM in water; Sigma–Aldrich), and an incubation period of 60 min at 37 °C followed. A control with sodium orthovanadate (1 mM; Sigma–Aldrich) was necessary for subtraction in the following calculations. The reaction was stopped by the addition of 5% SDS (Sigma–Aldrich). The samples were supplemented with Pi reagent (H_2_SO_4_ (2.5 M; Acros Organics, Geel, Belgium)), ammonium molybdate (1%; Sigma–Aldrich), antimony potassium tartrate (0.014%; Alfa Aesar), acetic acid (20%; Sigma–Aldrich), and freshly prepared ascorbic acid (1%; Sigma–Aldrich). After further 20 min incubation, the optical density was measured using a Paradigm^®^ microplate reader (Beckman Coulter, Germany) at a wavelength of 710 nm at room temperature. Calibration was accomplished with K_2_HPO_4_ (Sigma–Aldrich) and used to determine the amount of phosphate from the absorbance values. For the full-blown concentration-effect curve of DMS-SG, different concentrations of DMF (0.01 µM–5 mM) were added in different concentrations to a constant concentration of GSH (5 mM).

### 2.10. Multi-Drug Resistance Reversal Assay

The capability of DMF to reverse ABCC1-mediated MDR was determined by applying an MTT-based cell viability assay as described earlier [42]. The toxicity of the antineoplastic agent doxorubicin (0.01–10 µM; 20 µL/well; Calbiochem (Merck KGaA, Darmstadt, Germany)) toward ABCC1-expressing H69AR cells (20,000 cells/well; 160 µL/well) was assessed either alone or in combination with 5.0 µM, 10 µM, 20 µM, 30 µM, or 50 µM DMF (20 µL/well) and compared to the sensitive counterpart cell line H69 (20,000 cells/well; 180 µL/well). The necessary volumes were transferred onto clear 96-well flat-bottom plates, which were subsequently incubated for 72 h at 37 °C and 5% CO_2_-humidified atmosphere. Eventually, 40 µL of an MTT solution (5 mg/mL; Alfa Aesar) was added to each well, followed by incubation of 1 h. The supernatant was removed, and 100 µL of DMSO was added to each well. Absorbance measurement was spectrophotometrically accomplished at 570 nm using a Paradigm^®^ microplate reader (Beckman–Coulter, Germany; background correction: 690 nm). The determined absorbance values were plotted against the logarithmic concentrations of DMF, subsequently applying non-linear regression using Prism (v8.4.0, GraphPad Software, San Diego, CA, USA).

### 2.11. Cell Viability Assay

The intrinsic toxicity of DMF was assessed in a concentration range between 3.16 µM and 100 µM using the same MTT-based cell viability assay as described above.

### 2.12. Statistical Analysis

Statistical analysis was performed with Prism (GraphPad Software). We verified the data for Gaussian normal distribution by using the Shapiro–Wilk normality test. Student *t*-tests were performed to determine the significant differences between the two groups. Data are presented as means ± standard deviation (SD) or standard error of the mean (SEM). Differences were considered statistically significant when *p* < 0.05. N is reported in the figure legends.

## 3. Results

### 3.1. The Impact of DMF on Body Weight and Drinking Water Consumption

We treated APPtg mice with DMF through neutral drinking water (pH ~7.2) at an advanced stage of β-amyloidosis (125 to 175 days of age) at which animals already have abundant amounts of oligomeric Aβ and Aβ plaques in the brain [34,38,39].

First, we confirmed that DMF was stable in neutral drinking water (pH ~7.2) using spectroscopic measurements (Appendix B, Figure A1). The body weight of the mice was monitored weekly. As seen in Figure 3, DMF did not have a significant effect on body weight in male and female APPtg mice compared to controls despite a slight tendency of body weight loss in male late-stage amyloidosis mice.

In addition to body weight, we also monitored total weekly water consumption per cage and calculated the average amount of water consumed per animal per week. We found that DMF treatment considerably reduced water intake in male but not female APPtg mice (Figure 4). The intermittently reduced water consumption of male late-stage amyloidosis mice could be the reason for the slight tendency of reduced body weight.

### 3.2. The Impact of DMF on Aβ Deposition

To determine whether DMF treatment had an effect on late-stage β-amyloidosis mice, we determined cerebral levels of aggregated Aβ (higher MW aggregates, GuHCl extraction) as well as soluble Aβ (monomers and small oligomers, TBS extraction). We extracted total protein from the brain and first quantified aggregated Aβ42 in the guanidine-soluble fractions (Figure 5A,B). Similar to our previously published results in female younger APPtg animals with less advanced stages of β-amyloidosis, we did not detect changes in Aβ amounts after DMF treatment in female late-stage amyloidosis mice (Figure 5A). However, we observed a significant difference in aggregated Aβ42 in DMF-treated male APPtg mice compared to controls with lower Aβ42. This difference was not present in the TBS fraction that contains soluble Aβ42 (Figure 5C), pointing to no change in Aβ42 production and/or early deposition between DMF-treated and control animals. In essence, DMF seems to have no impact on Aβ deposition in late-stage amyloidosis mice.

### 3.3. The Effect of DMF on ABC Transporter Function

Previous research suggested that DMF may affect the function and regulation of ABC transporters [31,32]. Thus, we assessed the potential of DMF and its metabolite MMF to modulate the AD-associated ABC transporters ABCB1 (P-glycoprotein, P-gp), ABCC1, (multi-drug resistance-associated protein 1, MRP1), and ABCG2 (breast cancer resistance protein, BRCP1) in vitro. We applied two different assays per transporter to minimize the possibility of false-positive or false-negative outcomes, as functional assays strongly depend on the functional tracer used [43]. In principle, all assays use fluorescence dyes (or their precursors) that are substrates of the respective evaluated transporter. Inhibition of the transporter expressed in model cell lines results in increased intracellular concentrations of the fluorescence dyes (or their precursor), allowing for their spectroscopic determination. The higher the degree of inhibition of the used ABC transporter inhibitor, the higher the measured intracellular fluorescence values.

An ABCB1 modulation could not be observed as assessed in calcein AM and daunorubicin assays using ABCB1-expressing A2780/ADR cells, as already reported earlier [51] (Figure 6A). However, ABCC1 transport activity was inhibited by DMF at concentrations of 10 µM and 50 µM, respectively, as determined in daunorubicin and rhodamine 123 assays using ABCC1-expressing H69AR cells (Figure 6B). Finally, no modulatory effect could be observed against ABCG2 in both pheophorbide A and Hoechst 33,342 assays using ABCG2-expressing MDCK-II-BCRP1 cells (Figure 6C). In summary, we could show for the first time that DMF seems to functionally inhibit ABCC1-mediated transport of the ABCC1 substrates daunorubicin and rhodamine 123.

### 3.4. The Effect of DMF and Its Glutathione Conjugate on ABCC1 ATPase Activity

The observed inhibition of the ABCC1-mediated transport of both daunorubicin and rhodamine 123 prompted us to investigate whether DMF had an impact on the energy-supplying unit of ABCC1, the ABCC1 ATPase. As DMF has been demonstrated before to easily form glutathione conjugates under near-physiological conditions [52], we also analyzed its glutathione conjugate (dimethyl succinate-SG, DMS-SG, **9**; Figure 7A). As can be seen in Figure 7B, DMF alone had no effect on ATP cleavage (and phosphate liberation) mediated by the ABCC1 ATPase, while DMS-SG stimulated the ABCC1 ATPase to an even greater extent than the reference ABCC1 ATPase stimulator, reduced glutathione (GSH), alone (~1.6 times of GSH; Figure 7B). This effect was concentrat, ion-dependent, with a maximum at ~200 µM (Figure 7C). In contrast to inhibition, stimulation of the ATPase suggests that the stimulant is a substrate of the respective transporter [47]. This is particularly true with respect to GSH and GSH analogs, which were demonstrated earlier to be transported by ABCC1 under competitive inhibition of the ABCC1-mediated transport of daunorubicin [53,54,55]. Thus, the observed activation of the ABCC1 ATPase by DMS-SG only suggests that DMS-SG (and not DMF) is the actual effector in the functional assays, being itself a substrate rather than an inhibitor of ABCC1.

### 3.5. The Effect of DMF and DMS-SG on ABCC1 Transport Activity

The data shown in Figure 7 suggested that not DMF but its intrinsically formed GSH-adduct (DMS-SG) was transported by ABCC1, competing with daunorubicin and rhodamine 123 for the transport capacity of ABCC1. To prove this hypothesis, we evaluated both DMF and DMS-SG with respect to the concentration dependence of their effects and whether these concentration-effect curves exhibited differences. Strikingly, both compounds showed similar concentration-effect curves in both daunorubicin and rhodamine 123 assays with comparable half-maximal inhibition concentrations (IC_50_) values, allowing for the conclusion that DMS-SG is the actual effector against ABCC1 (Figure 8).

### 3.6. Efficacy of DMF in Cancer Cells Expressing ABCC1

In order to substantiate our findings, we sought to functionally assess the capability of DMF to reverse ABCC1-mediated multi-drug resistance (MDR) by applying a 3-(4, 5-dimethylthiazol-2-yl)-2, 5-diphenyltetrazolium bromide-(MTT)-based cell viability assay. ABCC1-expressing H69AR cells exhibit resistance against the antineoplastic agent doxorubicin. Inhibition of ABCC1 would sensitize H69AR cells, resulting in less doxorubicin necessary to impair H69AR cell viability. As can be seen from Figure 9A, DMF (5–30 µM) was able to shift the concentration-effect curve of doxorubicin from the right (resistant) to the left (less resistant) toward the sensitive cell line (no resistance). Concentrations higher than 30 µM could not be applied due to the occurring toxicity of DMF against H69AR cells. The half-maximal reversal concentration (EC_50_) was 8.81 µM (Figure 9B). This EC_50_ value is well in alignment with the IC_50_ values as determined in Figure 8, considering the usual discrepancy between inhibition and efficacy assays that is very often much greater than a factor of 2 [43,56]. The intrinsic toxicity of DMF can be visualized in Figure 9C; the determined half-maximal growth inhibition concentration (GI_50_) was 52.7 µM, which matches the findings of impaired cell viability at 50 µM as indicated in Figure 9A. In conclusion, DMF forms DMS-SG in vitro, which is a substrate of ABCC1 and competitively inhibits ABCC1-mediated doxorubicin transport, eventually reversing ABCC1-mediated MDR against doxorubicin in ABCC1-expressing H69AR cancer cells.

## 4. Discussion

In our previous work, we have shown that DMF had no effect on cognitive impairment, β-amyloidosis, and neuroinflammation in younger APPtg female animals (endpoints at 80 and 100 days, respectively) [3]. The aim of the present study was to complement the previous data and to investigate DMF treatment in older animals (endpoint at 175 days of age) as well as animals of both sexes, as our previous study was limited to females only. Another previous study by us using Fingolimod (FTY720), also a MS-repurposed drug, showed positive effects only in older male mice (treatment for 50 days from 125–175 days of age) [38].

To this end, we treated APPtg mice with DMF at a dose of 75 mg/kg body weight daily via drinking water. This dosage matched the one used in our previous experiments and was comparable to the dosage used in the treatment of multiple sclerosis patients (Data described in the pharmacological review application 204063Orig1s000 to the FDA by Paul C. Brown, (https://www.accessdata.fda.gov/drugsatfda_docs/nda/2013/204063Orig1s000PharmR.pdf (accessed on 10 May 2023)). Drug application through drinking water instead of a bolus application once a day was chosen to achieve a more even drug availability throughout the day, although it bore the risk of reduced drug intake due to low water consumption, which has been seen for male late-stage amyloidosis mice only (Figure 4B). However, although water and subsequent DMF uptake were documented for both female and male (slightly reduced) mice, DMF did not improve β-amyloidosis in agreement with our previous study [3].

Interestingly, a recent report suggested sex-specific effects of DMF on microglia exclusively in females [57], and another one addresses sex-specificity in AD mouse models [38]. While we have not investigated microglia in the present study, we did not observe microglial changes in females in our previous study [3]. Several research articles have been published (listed in Appendix A) using either transgenic mouse models [3,58] or experimentally inducible models in rats [28,29,30] (all from the same group) [59] and mice [60] to verify effects of DMF on the pathology and behavioral aspects also associated with AD. Inducible models using, e.g., streptozotocin with ICV delivery, have low construct validity since the underlying mechanism leading to disease-related effects is very different from the ones occurring in AD. This low predictive validity is expected to mask eventual positive effects for the treatment of AD and thus hampers the extrapolation from these induction experiments toward human AD patients. Apart from the induction models, only one publication proposed possible positive AD treatment effects of DMF. The authors used a double-transgenic mouse model (*APP* and *TAU*) and assessed activation of Nrf2-signaling by DMF, which led to a trend of reduced impairment of motor functions during the disease progress and improved memory through reduced neuroinflammation [58].

In the present study, we conducted additional comprehensive in vitro experiments to assess a potential direct effect of DMF and its metabolite MMF on ABC transporter activity, a group of transporters associated with AD [33,61,62,63,64]. As it has already been demonstrated in the literature, DMF (and MMF) did not promote or impede ABCB1 transport activity [51]. This also accounted for ABCG2, which up to now, has not been associated with DMF or MMF in the literature before. Strikingly, we could show inhibition of ABCC1 by DMF at concentrations of 10 µM and 50 µM. ABCC1 is an important gatekeeper at the blood–plexus barrier for Aβ clearance from the brain [34]. In vivo studies have previously shown that DMF reaches the brain only at low concentrations [65]. One reason could be a potential recognition and efflux mediated by ABCC1. In addition, DMF has been demonstrated to form glutathione conjugates [52], and many glutathione conjugates were demonstrated to be substrates of ABCC1 [53,54]. Thus, we have shown in the present study that DMS-SG, the GHS-conjugate of DMF, activated the ATPase of ABCC1, strongly suggesting promoted transport. Moreover, concentration-effect curves of both DMF and DMS-SG resulted in similar inhibitory effects, allowing for the conclusion that DMS-SG is the actual effector of ABCC1. Functional efficacy assays underpinned our findings, as DMF was able to partially reverse ABCC1-mediated MDR in ABCC1-expressing H69AR cells.

Although ABCC1 often co-transports substrates with GSH, this may be hindered in the case of Aβ protein due to the deprivation of ABCC1 transport capacity by DMS-SG. Comparable findings have already been found for other GSH-conjugates and daunorubicin transport [53,54,55]. These observations provide an explanation of why DMF was not able to ameliorate both early- and late-stage amyloidosis in APPtg mice in our previous [3] and the present study. Additionally, ABCC1 is expressed ubiquitously in the human body, including the intestine [66]. Importantly, the intestine is the site of uptake after oral application of DMF, leading to (i) short-term availability and (ii) higher concentrations of the compounds locally than in other parts. In this light, inhibition of ABCC1 may be of interest for further studies, especially given that the most common side effects of DMF treatment are gastrointestinal symptoms [67].

## 5. Conclusions

The present study addresses two important aspects of the potential use of DMF as a repurposed anti-AD drug. Firstly, we clarify the complete time frame for treatment with DMF on a late-stage β-amyloidosis mouse model. Secondly, we provide an explanation for the ineffectiveness of DMF in both early- and late-stage β-amyloidosis by thorough investigation of its interaction with the AD-related ABC transporter ABCC1. In light of the presented results, DMF becomes entirely disqualified as a repurposed drug to treat or ameliorate AD and AD progression.

## Figures and Tables

**Figure 1 biology-12-00932-f001:**
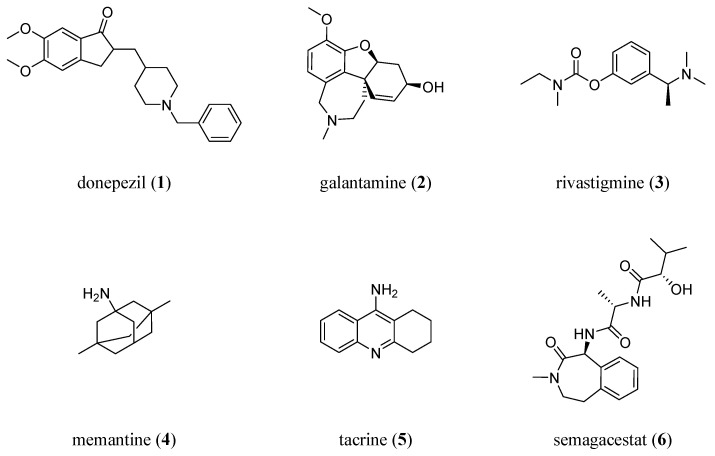
Molecular formulae of drugs to symptomatically or causatively address AD: donepezil (**1**; AChE inhibitor), galantamine (**2**; AChE inhibitor), rivastigmine (**3**; AChE inhibitor), memantine (**4**; NMDA antagonist), tacrine (**5**; first drug approved against AD; AChE inhibitor), and semagacestat (**6**; γ-secretases inhibitor).

**Figure 2 biology-12-00932-f002:**
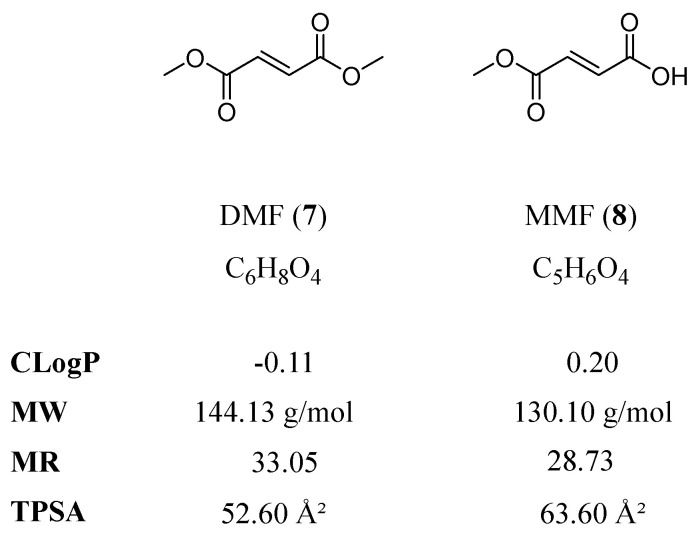
Molecular formulae of DMF and MMF, including the physicochemical parameters CLogP, MW, MR, and TPSA, are particularly important with respect to central nervous system (CNS) penetration and ABC transporters.

**Figure 3 biology-12-00932-f003:**
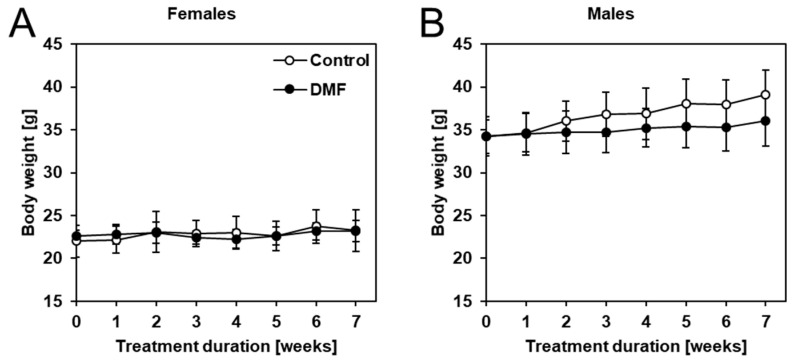
DMF treatment did not have a significant effect on the body weight of APPtg mice. Mice were weighed weekly, and no significant differences in body weight were observed comparing DMF-treated mice (closed circles) to controls (open circles) in (**A**) females and (**B**) males. Data are shown as mean ± standard deviation (SD); *n* = 7–8. Statistical analysis was performed using Student’s *t*-test.

**Figure 4 biology-12-00932-f004:**
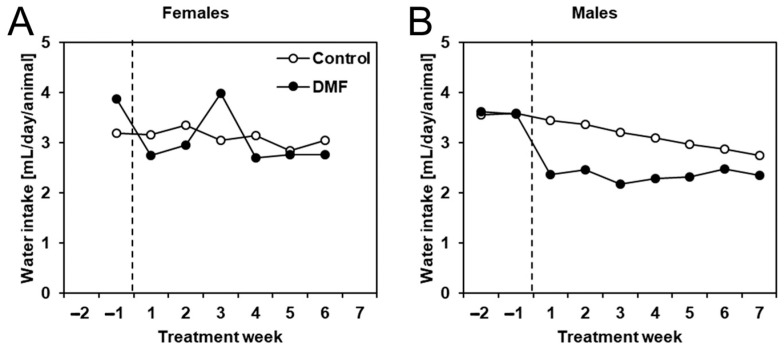
Male APPtg mice had reduced water intake while treated with DMF. Water consumption was measured per cage, and the weekly water intake was later calculated as mean water intake per animal (same for all animals, no SD) for (**A**) female and (**B**) male DMF-treated (closed circles, *n* = 8) and control APPtg animals (open circles, *n* = 7–8).

**Figure 5 biology-12-00932-f005:**
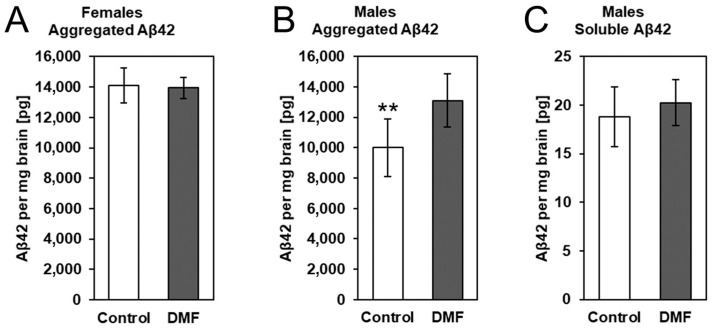
DMF had no effect on Aβ deposition in 175-day-old APPtg animals. We assessed Aβ levels in DMF-treated (gray bars) (**A**) female and (**B**) male APPtg mice compared to controls (white bars) using electrochemiluminescence-based immunoassays. Protein extraction from brain hemispheres was performed in two steps. Soluble Aβ42 was removed with a TBS buffer before extracting aggregated Aβ42 with GuHCl buffer. (**A**,**B**) Graphs show aggregated Aβ levels in control and DMF-treated (**A**) female and (**B**) male APPtg mice. (**C**) Soluble Aβ levels in control (white bar) and DMF-treated APPtg males (grey bar). Data are presented as mean ± SD; *n* = 6–8. Statistical analysis was performed with Student’s *t*-test. ** *p* < 0.01 considered significant.

**Figure 6 biology-12-00932-f006:**
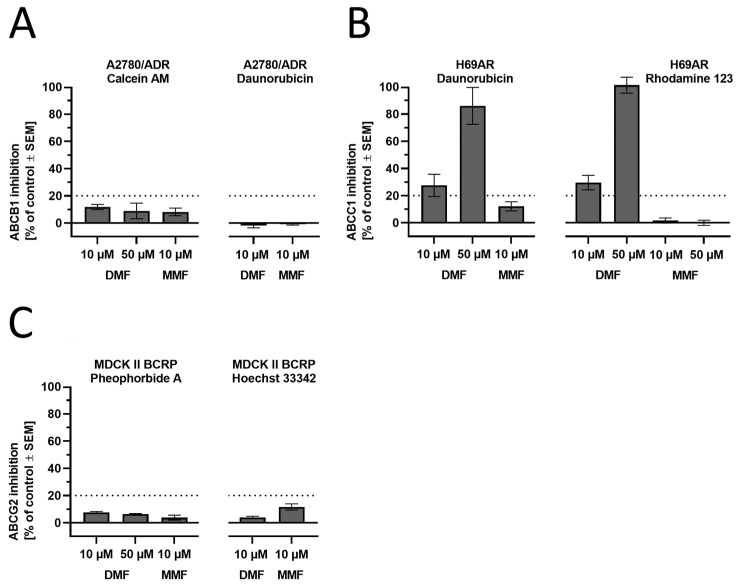
Functional screening of DMF and MMF against the AD-related ABC transporters (**A**) ABCB1, (**B**) ABCC1, and (**C**) ABCG2 applying calcein AM (ABCB1), daunorubicin (ABCB1 and ABCC1), rhodamine 123 (ABCC1), pheophorbide A (ABCG2), and Hoechst 33,342 (ABCG2) assays using ABCB1-expressing A2780/ADR, ABCC1-expressing H69AR, and ABCG2-expressing MDCK II BCRP cells. Full inhibition (100%) was determined by the effect of 10 µM of the reference inhibitors cyclosporine A (ABCB1), 4-chloro-6,7,8,9-tetrahydropyrimido [4,5-b]indolizine-10-carbonitrile (ABCC1) [46], and Ko143 (ABCG2). For all assays, the experimental cut-off for meaningful results was 20%, as visualized in the charts. Data are shown as mean ± standard error of the mean (SEM); *n* = 3–7.

**Figure 7 biology-12-00932-f007:**
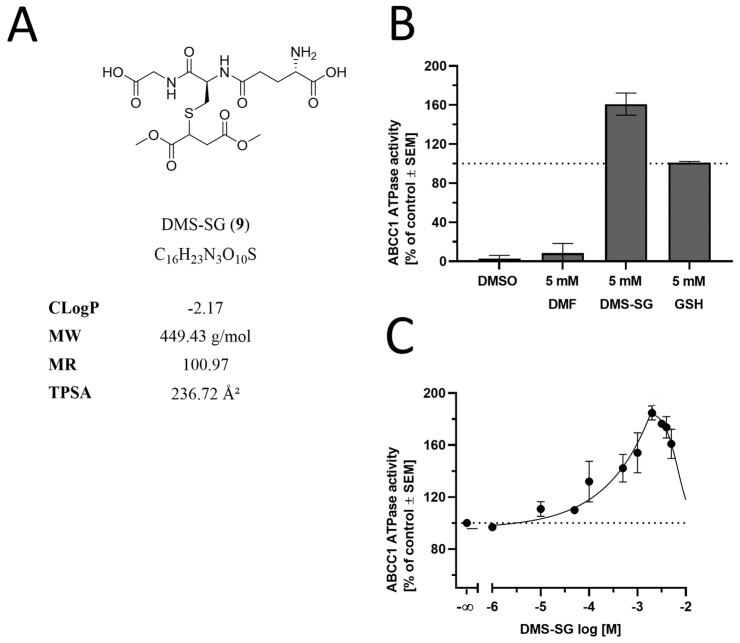
Effect of DMF, DMS-SG, and GSH on the ABCC1 ATPase. (**A**) Molecular formulae of DMS-SG, including the physicochemical parameters CLogP, MW, MR, and TPSA, are particularly important with respect to CNS penetration and ABC transporters [35]. (**B**) Screening of DMF (5 mM) and DMS-SG (5 mM) in a vanadate-sensitive ATPase assay using membrane preparations of ABCC1-expressing H69AR cells and colorimetric detection of liberated phosphate with ascorbic acid at 710 nm. Data are compared to the control (DMSO) and the reference ABCC1 stimulator GSH (5 mM) [53] and are shown as mean ± SEM; *n* = 3. (**C**) Concentration-dependent modulation of the ABCC1 ATPase by DMS-SG with maximal stimulation at ~200 µM. Data are shown as mean ± SEM; *n* = 3.

**Figure 8 biology-12-00932-f008:**
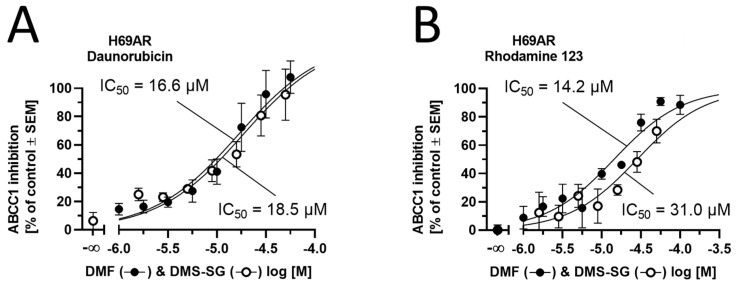
Concentration-effect curves of DMF (closed circles) and DMS-SG (open circles) obtained in the (**A**) daunorubicin and (**B**) rhodamine 123 assays using ABCC1-expressing H69AR cells. Data are shown as mean ± SEM; *n* = 3–4.

**Figure 9 biology-12-00932-f009:**
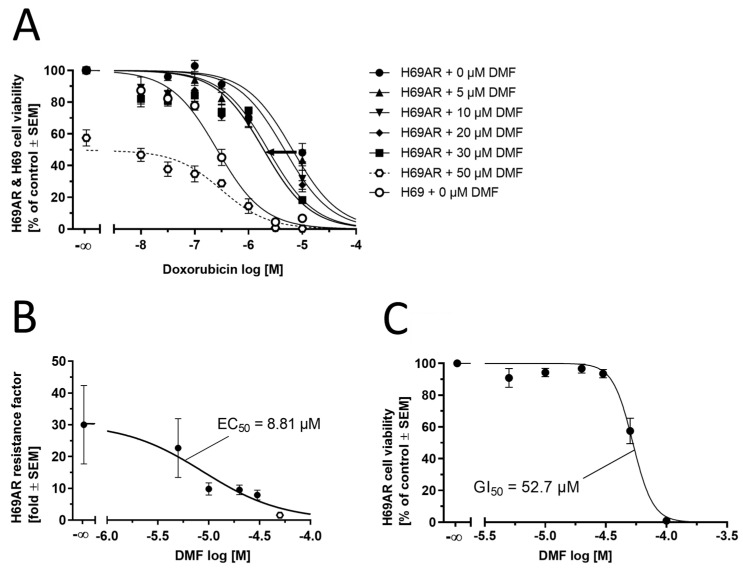
Complementary functional assessment of DMF against ABCC1. (**A**) Concentration-effect curves of the antineoplastic agent doxorubicin without (closed circles) and supplemented with 5.0 μM (closed upward triangles), 10 μM (closed downward triangles), 20 μM (closed routes), 30 μM (closed squares), and 50 µM (open hexagons and dashed line; outlier) of DMF as determined in an MTT-based cell viability assay using ABCC1-expressing H69AR cells compared to the concentration-effect curve of doxorubicin against the sensitive H69 cell line (open circles). (**B**) Plotting of the resistance factors (individual GI_50_ values of concentration-effect curves of doxorubicin in H69AR cells divided by the GI_50_ value of the concentration-effect curve of doxorubicin in H69 cells) allowed for the determination of the EC_50_ value of 8.81 μM (open hexagon: DMF at 50 µM not considered for non-linear regression). (**C**) Determination of intrinsic toxicity of DMF using H69AR cells. The GI_50_ value was determined to be 52.7 µM. Data are shown as mean ± SEM; *n* = 4.

## Data Availability

Data files and figures can be downloaded from the ABCS1P+ABCMS+5xFAD projects at DOI 10.17605/OSF.IO/VWQ58.

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
