# Peer review of "ABC Transporter C1 Prevents Dimethyl Fumarate from Targeting Alzheimer’s Disease"

_biology, 2023, doi:10.3390/biology12070932_

Round 1

Reviewer 1 Report

In this manuscript, the authors assessed the effect of dimethyl fumarate (DMF) on the treatment of Alzheimer’s disease (AD) using an APP-transgenic (APPtg) mice model. The authors identified several interesting findings. Meanwhile, there are a few concerns that may significantly weaken the manuscript.

1. In the results section, the authors didn't have subsections and the corresponding titles to highlight the key findings for each subsection. Please add accordingly.

2. Some paragraphs described the results, but didn't have conclusion sentences at the end of the paragraph. Please add.

3. I'm confused about the description of Figure 5.

1) "Similar to our previously published results in younger APPtg animals with less advanced states of β-amyloidosis, we did not detect changes in Aβ amounts after DMF treatment." I suppose this is the description of Figure 5A, which is the result from females. But the authors didn't mention female in the sentence.

2) We observed a significant difference of aggregated Aβ42 in DMF-treated male APPtg mice compared to controls. However, this difference was not present in the TBS fraction that contains soluble Aβ42 (Figure 5C), pointing to no change in Aβ42 production or deposition between DMF-treated and control animals. I'm confused about the conclusion. Based on Figure 5B and 5C, it showed increased aggregated Aβ42 in DMF group, but no difference in the TBS fraction (soluble Aβ42). What conclusion could be drawn? How many fractions are there? Aggregated Aβ42 in the guanidine-soluble fraction and soluble Aβ42 in the TBS fraction? What's the biological meaning of the two fractions? Please provide more information in the beginning of the paragraph.

4. Based on the figure 3, 4 and 5, it showed the DMF treatment in male APPtg mice led to decreased bogy weight trend (although it's not significant), reduced water intake and increased aggregated Aβ42. These results indicated that the drug DMF showed no beneficial effect, but deteriorative effect in the treatment of male APPtg mice. How to rationale this effect?

5. When assessing the modulation role of DMF/MMF on ABCB1/ABCC1/ABCG2, why the authors used different assays? Please explain.

6. In the abstract, the authors mentioned "glutathione-modified DMF interacts with the ATP-binding 42 cassette transporter ABCC1". I'm not sure which figure supports this statement. Figure 7 showed the effect of DMS-SG (glutathione-modified DMF) on the ABCC1 ATPase activity. Figure 8 showed similar effect of DMF and DMS-SG on ABCC1 inhibition. The authors used the word "interacts", does it refer to the physical interaction between DMS-SG and ABCC1? Please clarify.

7. "The half-maximal reversal concentration (EC50) was 8.81 μM (Figure 9B), which is in alignment with the IC50 values as determined in Figure 8." In this statement, the authors showed EC50 is 8.81 μM, but in Figure 8, the IC50 for DMF is 16.6 μM in daunorubicin assay, and 14.2 μM in Rhodamine 123 assay. How the authors draw the conclusion that the two results are in alignment. Please provide more detailed explanation.

The quality of English language looks good.

Author Response

see attached pdf

Reviewer 2 Report

The authors of «  ABC Transporter C1 Prevents Dimethyl Fumarate from Targeting A_l_z_h_e_i_m_e_r_’s_ _Disease » treat old animals of an Alzheimer disease model, with  dimethyl fumarate (DMF), an already approved for humans medicament and used in different diseases , as psoriasis or multiple sclerosis. This paper is the continuation of the previous research made by the same group where they tested DMF on young animals of the same AD model with no results. 

I have to say that it seems really brave to engage in this procedure since the expected result was a negative result, as they obtained, but they go further to try to find an explanation for this lack of effectiveness. Based on the literature, they checked if DMF or its bioactive metabolite affected the correct function of several ABC transporters, finding that DMF is able to inhibit the ABCC1 transporter on cultured cells expressing this transporter, especially if it was conjugated with a glutathione group. 

Having said that, I think the title misguides the possible reader. I understand they want to link the transporter, Alzheimer disease and DMF, but after reading the paper my impression is that DMF is inhibiting ABCC1, and then it is not able to cross the BBB and/or it doesn’t help to the clearance of amyloid beta from the cerebrospinal fluid surrounding the neurons. Then, DMF it is not suitable as treatment, it should worsen Abeta deposition, what actually happens in male treated animals. I think they should make clearer their hypothesis and a more comprehensible title.

In addition, I have several other points that should be clarified.

 In Material and Methods, line 318 they said they are using“Heterozygous female and male APPPS1-21 mice », why? Instead of using homozygous of a stable animal line, why do they use heterozygous? I don’t say any advantage, could they explain it? 

In the same section, line 323, “and acidified water (pH 3) », that was really surprising, and my only explanation it is is that it was a mistake, specially since later in line 335 & 341 they used “in drinking water (pH ~7.2) » and « water at neutral 341 pH ».

Also, a question, since they fixed an hemisphere, line 357, why they didn’t check the number and localization of Abeta plaques? 

Figure 3, I don’t think it will change the non-significant result, but since they are the same animals followed through several weeks, the correct statistical test should be a repeated measures t-test. 

Line 136 and Figure 4, what statistical test was done to affirm « reduced water intake in male »? Why there are no error bars? The cited n is number of cages? Or of animals in one cage? Because then it will hard to extract any conclusion from just one measure.

Line 150, I don’t think this phrase is correct “A_β4_2_ _production or 150 deposition between DMF-treated »since in line 148 they claim “We observed a significant difference of a_g_g_r_e_g_a_t_e_d_ _A_β4_2_ _i_n_ _D_M_F_-treated male AP-148 Ptg mice compared to controls ».  Indeed, They do have an effect, just opposite to the expected one. 

Figure 5, the difference would be more clear if they would represent mean + SEM instead of mean ± SD, as they do in the rest of the figures. 

Paragraph starting in line 160, it should be explained better. There are many assumptions that probably most of the non-specialized readers are not able to understand. Since the authors changed to an in vitro system, they should explain it and what exactly are they measuring.   

Line 186, figure 7, it is unclear why they decided to test DMS-SG, until it is discussed later in line 201. It will be clearer for the reader if it will be presented first, understanding how the experiments were planned.

Figure 9 (and also 8), they will win clarity if they would include a legend, showing the meaning of the symbols, concentrations…

Figure A1, on C the linear correlation is lost. Does it form aggregates or does it precipitate? What’s their explanation? On D, n should achieve 3 at least. 

So, even if I think it was courageous to perform the research until the end, I think that there are too many points to clarify,  and some parts of the text should be reformulated to be presented in easier way for the non-specialized reader. 

Author Response

see attched pdf

Round 2

Reviewer 2 Report

The authors decided to discuss all the suggestions and apply all the possible changes without modifying neither the experiments nor the graphs.

I think the paper is more clear and easier to understand. I still think that the proper way to do it is increasing the “n”, different stats and the points already discussed, but since I don’t think that these changes will change their conclusion, I will not oppose to the publication of the paper in the present form.